# Iron Deposition and Ferroptosis in the Spleen in a Murine Model of Acute Radiation Syndrome

**DOI:** 10.3390/ijms231911029

**Published:** 2022-09-20

**Authors:** W. Bradley Rittase, John E. Slaven, Yuichiro J. Suzuki, Jeannie M. Muir, Sang-Ho Lee, Milan Rusnak, Grace V. Brehm, Dmitry T. Bradfield, Aviva J. Symes, Regina M. Day

**Affiliations:** 1Department of Pharmacology and Molecular Therapeutics, Uniformed Services University of the Health Sciences, Bethesda, MD 20814, USA; 2Department of Pharmacology and Physiology, Georgetown University Medical Center, Washington, DC 20007, USA; 3Department of Pathology, Uniformed Services University of the Health Sciences, Bethesda, MD 20814, USA; 4Department of Laboratory Animal Research, Uniformed Services University of the Health Sciences, Bethesda, MD 20814, USA

**Keywords:** radiation, spleen, iron, ferroptosis, macrophage

## Abstract

Total body irradiation (TBI) can result in death associated with hematopoietic insufficiency. Although radiation causes apoptosis of white blood cells, red blood cells (RBC) undergo hemolysis due to hemoglobin denaturation. RBC lysis post-irradiation results in the release of iron into the plasma, producing a secondary toxic event. We investigated radiation-induced iron in the spleens of mice following TBI and the effects of the radiation mitigator captopril. RBC and hematocrit were reduced ~7 days (nadir ~14 days) post-TBI. Prussian blue staining revealed increased splenic Fe^3+^ and altered expression of iron binding and transport proteins, determined by qPCR, western blotting, and immunohistochemistry. Captopril did not affect iron deposition in the spleen or modulate iron-binding proteins. Caspase-3 was activated after ~7–14 days, indicating apoptosis had occurred. We also identified markers of iron-dependent apoptosis known as ferroptosis. The p21/Waf1 accelerated senescence marker was not upregulated. Macrophage inflammation is an effect of TBI. We investigated the effects of radiation and Fe^3+^ on the J774A.1 murine macrophage cell line. Radiation induced p21/Waf1 and ferritin, but not caspase-3, after ~24 h. Radiation ± iron upregulated several markers of pro-inflammatory M1 polarization; radiation with iron also upregulated a marker of anti-inflammatory M2 polarization. Our data indicate that following TBI, iron accumulates in the spleen where it regulates iron-binding proteins and triggers apoptosis and possible ferroptosis.

## 1. Introduction

The hematopoietic system is markedly sensitive to damage by radiation [1]. In humans, exposure to 4–6 Gy of total body irradiation (TBI) can lead to the development of hematopoietic subsyndrome of acute radiation syndrome (H-ARS), characterized by hematopoietic insufficiency, opportunistic infection from immune suppression, acute inflammatory response, and coagulation dysfunction [1,2,3,4,5]. Mature white blood cells (WBC) and many hematopoietic progenitors undergo apoptosis following exposure to high-dose radiation [5,6]. Reticulocytes and red blood cells (RBC) lack DNA and apoptotic machinery and instead undergo hemolysis driven by the oxidation and denaturation of hemoglobin (HGB) [5,7,8,9]. Apoptosis of mature WBC contributes to immune insufficiency and opportunistic infection [10], while the destruction of RBC contributes to poor oxygenation and the release of potentially toxic iron [9,11].

Iron is a nutritionally essential element, required for many basic biological activities, including the generation of ATP by mitochondrial respiration and oxygen transport in the blood [12]. The conversion of iron between the ferric (Fe^3+^) and ferrous (Fe^2+^) states allow single electron transfers, a process required in a variety of biochemical reactions [12]. However, excessive iron is associated with biologically toxic effects. Free iron can promote the generation of toxic free radicals, as described in the Fenton and Haber-Weiss reactions [13,14]. Dysregulated iron accumulation in a variety of tissues and iron-catalyzed free radicals has been proposed to play a role in cancer, metabolic diseases, blood disorders, and multiple degenerative diseases [14,15,16]. 

Proteins involved in the binding and transport of iron are highly conserved in mammals [17]. Additionally, the processes for iron handling, from absorption in the intestine to safe transport in the plasma and storage in the liver, are tightly regulated and highly conserved evolutionarily [17,18,19,20,21]. Erythropoiesis accounts for the most abundant usage of iron, and 65–75% of the total iron in mammals is bound to hemoglobin within RBC [17]. Homeostatic iron recycling occurs within the spleen, where RBC are removed from circulation based on abnormal size, shape, deformability, and markers of senescence [22]. Effete or damaged RBC, detected within the spleen, undergo erythrophagocytosis by specialized red pulp macrophages [23,24]. In other tissues, macrophages provide iron homeostasis by sensing and responding to microenvironmental iron concentrations and taking up excess iron [25]. Although specialized macrophages normally take up iron for recycling through erythrophagocytosis, massive hemolysis with excessive iron release can suppress the phagocytosing function of macrophages, possibly preventing toxic overload [23]. The effects of iron on macrophage function and polarity appear to be complex. Several studies showed that iron accumulation in macrophages interfered with antimicrobial activity and nitric oxide production [25,26]. Studies also showed that in some cases iron supplementation promoted the expression of pro-inflammatory (M1) markers and reduced the expression of anti-inflammatory (M2) markers, while other studies showed the promotion of M2-like macrophage polarization [25,26]. This suggests that the response of macrophages to iron may be context-dependent.

Our laboratory and others have demonstrated that radiation-induced hemolysis of RBC is associated with an accumulation of iron in the bone marrow in rodent models of H-ARS [27,28]. The reduction of circulating RBC increased iron in the plasma, and increased iron levels in hemosiderin in the bone marrow have been associated with increased levels of hepcidin expression in the liver following TBI [28,29]. This suggests that iron release from intestinal cells is suppressed in response to high plasma iron following TBI. Here we investigated the deposition of iron in the spleen following TBI in mice. We found that Fe^3+^ iron accumulation in the spleen was maximal ~14 days post-irradiation, correlating with the maximal loss of RBCs and hematocrit (HCT) after TBI. Increased iron in splenic tissue was associated with increased levels of iron binding and transport proteins. Increased iron in the spleen was associated with the activation of caspase-3, a regulator of programmed cell death. Further investigation revealed the simultaneous regulation of several markers for ferroptosis, an iron-dependent form of programmed cell death. Histological analysis indicated that iron storage and expression of iron storage proteins within the spleen occurred mostly within splenic macrophages. As iron and radiation are known to affect the polarization of macrophages, we investigated the effects of iron, radiation, and iron + radiation on M1 and M2 polarization of J774A.1 murine macrophages in culture. Exposure to radiation induced accelerated senescence in cultured macrophages. Radiation and radiation + iron resulted in the upregulation of markers of M1 macrophages. Radiation + iron moderately upregulated one marker of M2 polarization in the cultured macrophages. These data provide evidence for Fe^3+^ release as a secondary toxic event following exposure to TBI. Additionally, the presence of Fe^3+^ following radiation exposure in vivo may play a role in pro-inflammatory responses after TBI.

## 2. Results

### 2.1. Sub-Lethal Irradiation Reduces Red Blood Cells, Hematocrit, and Hemoglobin 

Female C57Bl/6J mice were exposed to 6.85 Gy TBI (0.6 Gy/min). This sublethal dose of radiation did not result in mortality in either vehicle- or captopril-treated irradiated groups. Previous studies showed that radiation induces the hemolysis of RBC and reticulocytes due to the denaturation of hemoglobin (HBG) [8,9]. 6.85 Gy irradiation caused a significant reduction in RBC, HCT, and HGB, with maximal ~60% reduction around day 14 (Figure 1A–C). 6.85 Gy exposure caused less blood cell level reduction compared with a 50–70% lethal dose of radiation (7.75–7.9 Gy TBI) [30], and no significant differences were observed between vehicle- and captopril-treated irradiated animals (Figure 1A–C). The mean corpuscular volume (MCV), an indicator of the average RBC size, was near baseline on days 7 and 14 post-irradiation in both vehicle and captopril groups (Figure 1D). MCV exceeded the sham control volumes on day 21 in vehicle-treated animals, but not in the captopril-treated group (Figure 1D). The mean corpuscular HGB concentrations (MCHC) were maintained in captopril-treated irradiated mice over the full duration of the time course, while in vehicle-treated animals, the MCHC fell below basal levels at day 21 post-irradiation (Figure 1E). We also observed a 95–98% reduction in reticulocyte levels at 7 days post-TBI in both vehicle and captopril treated, irradiated groups. Reticulocyte levels returned to baseline by 14 days, and at 21 days there was a ~14-fold increase in reticulocytes in vehicle-treated irradiated animals and ~9-fold increase in reticulocytes in captopril-treated irradiated animals (Figure 1F). 

MCV (average erythrocyte size) and MCHC are two basic hematological changes associated with reticulocytosis. An increased MCV and decreased MCHC can occur due to partially retained ribosomal RNA material in prematurely released erythrocytes during regenerative anemia [31]. Together, the data suggest that in the vehicle-treated group the increased MCV and decreased MCHC may be the result of rapid erythropoiesis following acute erythrocyte hemolysis from TBI. This represents an erythropoietic response to the loss of RBC, with the loss most pronounced at 14 days. The erythropoietic response was most noticeable on day 21 for the vehicle-treated group as indicated by the maximum MCV suggesting a robust release of immature RBC from the bone marrow in the forms of larger reticulocytes to compensate for the radiation-induced loss of RBC.

We also observed transient suppression of monocytes following 6.85 Gy TBI (Figure 1G,H). Monocytes were suppressed by ~80–90% and 90–95% in vehicle-treated, and captopril-treated irradiated animals, respectively, at 7–14 days post-irradiation compared with sham irradiated control levels (Figure 1G). Monocytes recovered in both groups by 21 days and were above baseline levels at 28 days, with a significant increase in captopril-treated irradiated animals compared with vehicle-treated irradiated animals. In contrast, neutrophils did not fall significantly below the control levels (Figure 1H). Neutrophils were above the control levels at 28 days post-irradiation in the captopril-treated irradiated group (*p* < 0.05). 

### 2.2. Sub-Lethal Irradiation Results in Iron Deposition in the Spleen and Upregulation of Genes Encoding Iron Handling Proteins

Our laboratory and others demonstrated that following radiation-induced hemolysis, iron is deposited within the bone marrow [28,29]. However, iron recycling normally occurs within the spleen, where specialized macrophages take up senescent or damaged RBC [24]. We therefore investigated spleen iron levels following TBI. Within 7–14 days, Prussian blue staining, which detects Fe^3+^, increased ~20–60-fold (Figure 2A,B). Prussian blue staining returned to near basal levels by day 21. Captopril treatment did not significantly alter the time course or pattern of iron deposition in the spleen after radiation (Figure 2B; Appendix A).

Import of iron by specialized macrophages can occur through erythrophagocytosis (ingestion of whole RBCs) and by specific receptors, including transferrin receptor 1, lipocalin-2, and integrin alphaM [23,25,32,33]. Additionally, the feline leukemia virus subgroup C receptor can be utilized for iron export from macrophages and other cell types [23,25]. Ferritin is the common protein for protein storage within macrophages and some other cell types, and transferrin is normally used for the transport of ferric iron through the plasma [23,25]. It is important to note that integrin alphaM is also recognized as a general marker for macrophages, so regulation of this gene may also reflect the general population of macrophages. We examined gene expression of iron binding and transport proteins following exposure to TBI (Figure 3). Three patterns of gene expression changes were evident in the data. In the first group, *Fth1* (ferritin heavy chain) and *Trf* (transferrin) were significantly upregulated at 7 days, with declining expression to near basal levels at 14–28 days post-irradiation (Figure 3A,B). The second group, *Tfrc* (CD71/transferrin receptor), *Itgam* (integrin alphaM, also known as Mac-1 or CD11b/CD18), and *Lcn2* (lipocalin-2) exhibited initial suppression followed by increased expression (Figure 3C–E). Increased expression of both *Itgam* and *Lcn2* was especially noted in the captopril-treated irradiated group at day 21, in advance of the expression in the vehicle-treated irradiated group which increased at day 28 post-irradiation. The CD71/transferrin receptor binds transferrin from the plasma for the import of iron into the cell, while integrin alphaM has been shown to bind and import iron oxide nanoparticles in monocytes [33,34]. Lipocalin-2 binds iron for uptake into cells, including macrophages, where it can be sequestered [35,36,37]. Finally, we examined the expression of *Flvcr1* which encodes two heme export proteins important for cellular heme homeostasis: feline leukemia virus subgroup C receptor 1a (FLVCR1a) a plasma membrane heme exporter, and FLVCR1b, a mitochondrial protein [38,39]. *Flvcr1* displayed reduced expression (~50%) at 28 days in vehicle-treated animals (Figure 3F), suggesting a reduction in the uptake of heme-bound iron. In the irradiated animals, captopril treatment did not alter the expression of most of these genes. Captopril enhanced the expression of integrin alphaM at 21 days, with a trend toward increased expression at 28 days, and captopril enhanced the expression of lipocalin-2 at 21 and 28 days (Figure 3D,E).

Iron is exported by cells, including erythrophagocytosing macrophages and enterocytes, through the transporter ferroportin-1 that passes cellular iron to transferrin, the primary protein for the secure transport of iron through the plasma [23]. *Slc40a1* (ferroportin-1) expression was significantly upregulated at 7 days, with declining expression to near basal levels at 14–28 days post-irradiation (Figure 4A). However, ferroportin protein stability and its localization at the cellular membrane are additionally regulated by other factors, such as hepcidin, which suppresses iron export in reticuloendothelial macrophages and duodenal enterocytes [40]. We previously showed that hepcidin gene expression is upregulated in the liver at 7 days post-irradiation [28]. We observed a trend toward decreased ferroportin protein post-irradiation in vehicle- or captopril-treated animals, although this did not reach significance (Figure 4B). Thus, although ferroportin gene expression is observed to increase, ferroportin protein levels are not increased, potentially due to downregulation by hepcidin.

Immunohistochemistry (IHC) was used to identify cell types expressing several of the iron handling proteins. As described above, ferritin gene expression increased at ~7 days (Figure 5A,B). IHC showed a trend toward increased ferritin expression in the spleen at 7 days, declining at 14 and 21 days. Ferritin staining in irradiated spleens was granular in appearance and localized to the cytoplasm and dendritic processes of histiocytes/macrophages, with especially strong staining in cells in the sinuses under the spleen capsule (Figure 5A). Western blotting showed significant increases in ferritin (~5–6-fold, *p* < 0.05) at 7–14 days (Figure 5C). Hemosiderin, an iron storage complex containing ferritin and Fe^3+^, appeared as a gold-brown coloration (Figure 5D). Data show that there was a significant increase in hemosiderin at 7–14 days (~4-fold higher than control, *p* < 0.05), and 28 days (~2-fold higher than control, *p* < 0.05). Higher power magnification images for cell identification are shown in Appendix A.

Previous studies showed that the CD71/transferrin receptor is primarily expressed in early-stage erythroblasts, with little or no expression in mature RBC [24,32,41]. However, erythropoietic stress (for example after phlebotomy), can result in a population of transferrin-positive RBC [42]. In our experiments, qPCR showed an increase in the mRNA for the CD71/transferrin receptor, ~14–21 days post-irradiation. IHC showed a trend toward increased levels (~2-fold) at 21 days post-irradiation (Figure 6A,B), with prominent staining in the red pulp, localized to the cytoplasmic membranes of erythroid-like cells. This included nucleated immature RBC and cells that appeared to be mature RBC. A higher magnification image is shown in Appendix A. Reticulocytes were histologically characterized by a slightly larger diameter with a pale bluish tinge dispersed in the cytoplasm as compared to mature RBC [31]. The precursors to reticulocytes, metarubricytes, were identified by distinct darkly basophilic round nuclei [31] (Appendix A). The presence of CD71/transferrin receptor staining in mature RBCs in the spleen after TBI may be an indication of stress erythropoiesis following TBI [42]. CD71/transferrin receptor staining was not identified within histiocytes/macrophages. A high-powered magnification image is shown in Appendix A. Western blotting confirmed ~40-fold upregulation of CD71/transferrin receptor at 7–14 days (*p* < 0.05) (Figure 6C). 

### 2.3. Reduction in Spleen Weight Post-Irradiation Is Associated with Apoptosis and Possible Ferroptosis

Our laboratory previously showed that the area of the spleen was significantly reduced at ~10 days following TBI in mice [43]. We investigated the effect of radiation on the spleen weight, normalized to total body weight (Figure 7A). The normalized spleen weight was reduced to ~50% of control (*p* < 0.05) at 7- and 14-days post-irradiation in both vehicle and captopril-treated animals, correlating with the times of highest levels of iron deposition in the spleen. We performed western blotting for proteolytic activation of caspase-3, as a marker of programmed cell death [44]. Western blotting revealed a significant increase in caspase-3 proteolytic fragments at 7- and 14-days post-irradiation in vehicle-treated mice (Figure 7B, left panel; *p* < 0.05). In captopril-treated animals, caspase-3 activation also increased significantly at 14 days post-irradiation (Figure 7B, right panel; *p* < 0.05). We did not observe increased expression of p21/Waf1, a marker of cell cycle inhibition and accelerated senescence in the spleen after TBI (Appendix A). 

Recent studies showed that high levels of iron can induce ferroptosis, a form of iron-dependent programmed cell death [45,46]. Ferroptosis markers include increased expression of CD71/transferrin receptor 1, reduced glutathione peroxidase 4 (GPX4,) and solute carrier family 7 member 11 (Slc7A11) [45,47]. We demonstrated increased expression of the CD71 using qPCR (Figure 3C), IHC, and western blotting (Figure 6); IHC of the spleen showed that this increase occurred within mature and immature RBCs. We next investigated the gene expression of *Gpx4* and *Slc7a11* in the total spleen tissue (Figure 7C,D). *Gpx4* expression showed a trend toward decreased expression (~25% reduced) within 7 days post-irradiation, and was significantly suppressed, to less than 50% basal levels, at 14- and 21-days post-irradiation (*p* < 0.001–0.0001) compared with basal levels. In vehicle-treated animals *Gpx4* returned to near basal levels at 28 days post-irradiation; captopril treatment resulted in *Gpx4* remaining significantly lower than basal levels at 28 days (*p* < 0.05). In contrast, *Slc7a11* showed a slight trend toward increased expression at 7 days post-irradiation, followed by a trend toward reduced expression at 14 days in both vehicle- and captopril-treated irradiated groups. *Slc7a11* was significantly reduced only in vehicle-treated animals at 21 days post-irradiation (*p* < 0.0001). In captopril-treated animals, *Slc7a11* remained near basal levels over the time course. Cyclooxygenase-2 (COX-2) is also a marker, but not a driver, of ferroptosis [45]. Western blotting showed that COX-2 protein levels were significantly increased at 7 days post-irradiation in vehicle-treated animals (Figure 7E, left panel; *p* < 0.05). We observed a trend toward increased COX-2 protein in captopril-treated irradiated animals, but this did not reach significance (Figure 7E, right panel). 

To ensure that iron was indeed increased in the samples following total body irradiation, iron was directly assayed in the spleen tissues. Data show that there was a trend toward increased iron at 7 days post-irradiation, and iron was significantly increased at 14 days post-irradiation (Figure 7F). Lipid oxidation is a characteristic of ferroptosis, including malondiadehyde (MDA) [45]. We observed a trend toward increased protein-linked MDA at 7 days post-irradiation that was significant at 14 days post-irradiation (Figure 7G). The increase in iron and protein-linked MDA correlate with the 7–14 daytime point for increased apoptosis.

### 2.4. Alterations in M1 and M2 Gene Expression In Vivo and in Culture Murine Macrophages by Iron and Radiation

Macrophages play a key role in iron homeostasis as well as in normal immune responses [23]. Iron regulates iron binding and transport proteins in macrophages and modulates macrophage polarity [17,48,49]. Iron exposure upregulates ferritin and several other iron storage and transport proteins in macrophages [17], and impairs the ability of macrophages to assume a full pro-inflammatory (M1) phenotype [48]. However, in vitro studies showed that radiation had no effects on macrophage polarity [50]. In contrast with in vitro studies, in vivo studies of TBI have shown effects on macrophage polarization, favoring the induction of M1 polarization early after radiation, followed by the development of alternatively activated (anti-inflammatory or M2) macrophages [51,52]. 

We examined the expression levels of *Cd206*, a marker of M2 macrophage activation, and the expression of *Cd80*, a marker of M1 macrophage activation in mouse spleens after radiation (Figure 8). Our gene expression data showed a ~5-fold increase in *Cd206* and ~2.5-fold increase in *Cd80* expression at 7 days post-irradiation in both vehicle- and captopril-treated irradiated mice. The elevated levels of *Cd206* expression at 7 days post-irradiation is the same time we observed downregulation of the iron-binding protein and macrophage marker integrin alphaM, suggesting that increased upregulation of these markers is not due to increased macrophage populations at this time point. At 21 days post-irradiation, *Cd206* and *Cd80* were decreased to only 15% of sham-irradiated control levels in the vehicle-treated irradiated animals. In captopril-treated irradiated animals, *Cd206* and *Cd80* remained near control sham irradiated levels of expression. It is important to note that CD80 is also expressed on some B cells, T cells, dendritic cells, and monocytes. These data are suggestive of changes in M1 and M2 macrophage polarization, but specific determination of M1 and M2 macrophage polarization levels in vivo requires flow cytometry studies.

As the in vivo effects of total body radiation likely reflect the impact of radiation with iron on macrophages, we wished to determine the effects of iron, radiation, and iron + radiation on macrophage polarization in vitro. We utilized 7 and 12.5 mg/L Fe^3+^ in the medium based on reported iron concentrations in the serum following TBI in mice [9]; iron was combined with two doses of radiation that are sublethal when used in TBI studies in C57BL/6 mice [30]. We examined the gene expression of three markers of pro-inflammatory M1 polarization, inducible nitric oxide synthase (iNOS), interleukin-6 (IL-6,) and IL-1β (Figure 9A–C). Of these, radiation significantly induced *Nos2* and *Il1b*, but not *Il6*. Iron alone did not activate any markers of M1 polarity, but the presence of iron in the medium did show a trend toward augmenting iNOS expression (Figure 9A). Neither iron alone nor radiation alone upregulated the M2 marker *Arg1* (arginase 1) (Figure 9D). Iron + radiation did result in the upregulation of *Arg1*, although not consistent for all radiation + iron conditions (Figure 9D). We next examined the ability of M1- or M2-inducing cytokines (IFN-γ + LPS for M1; IL-4 for M2) to induce polarization in the presence of iron, radiation, or iron + radiation (Figure 10). Iron, radiation, or iron + radiation did not inhibit the upregulation of *Il6* or *Il1b* induction by IFN-γ + LPS or *Arg1* induction by IL-4. Together, these data suggest that radiation and iron + radiation may induce mixed polarity, but they appear to block cytokine responses by the macrophages in culture. 

Our in vivo data indicated that macrophages in the spleen contain high levels of ferritin after TBI. To determine the effects of high ionizing radiation and iron concentrations on iron binding protein expression, we examined the ferritin levels in J774A.1 cultured macrophages. We found ~6–14-fold upregulation of ferritin heavy chain in all cultures with high iron within 24 h (*p* < 0.05, Figure 11A). Interestingly, upregulation of ferritin heavy chain protein was not significantly affected by radiation exposure or by M1- or M2-inducing cytokines, although radiation and M1-inducing cytokines did upregulate gene expression of *Fth* at 24 h (Appendix A). We did not detect significant levels of transferrin receptor 1 in the cultured macrophages before or after irradiation (data not shown). 

In vivo data also indicated that significant levels of programmed cell death, but not accelerated senescence, occurred in the spleen following TBI. However, our previous studies indicated that ionizing radiation induces significant levels of accelerated senescence in cultured pulmonary artery endothelial cells [53]. The spleen contains significant levels of macrophages as well as microvascular endothelial cells, so we examined iron and radiation effects on cell death and senescence in cultured J774A.1 and human spleen microvascular endothelial cells. J774A.1 cells displayed a trend toward increased p21/waf1, a marker for accelerated senescence, after exposure to 2 or 6.85 Gy ionizing radiation (Figure 11B). Senescence was significant in 6.85 Gy irradiated cells in the presence of iron (Figure 11B). Neither iron alone nor M1- or M2-inducing cytokines induced p21/waf1. We did not detect significant caspase-3 activation under any conditions (data not shown). Examination of the effects of radiation and high iron on human spleen microvascular endothelial cells (HSMVEC) in culture showed neither caspase-3 activation nor p21/waf1 upregulation at 24 h following radiation, iron, or iron + radiation exposure (Appendix A). 

## 3. Discussion

Exposure to total body ionizing radiation can result in H-ARS, characterized by hematopoietic insufficiency, immune suppression, opportunistic infection, and coagulopathy. The acute effects of radiation have been extensively studied in vivo and in vitro, but the etiology of some of the chronic effects of radiation (weeks or months after initial exposure) is not well understood. Here we demonstrate that exposure to sublethal TBI results in the loss of RBC and HGB, allowing the release of iron that can be sequestered within the spleen, detected as Fe^3+^ complexes. Our data indicate that deposition of iron in the spleen is correlated with the regulation of iron binding and transport proteins and with apoptosis and possible ferroptosis. This suggests that iron, released from radiation-induced hemolysis, may act as a secondary toxic agent following exposure to even sublethal levels of ionizing radiation.

Our data indicated that there are at least three phases of regulation of iron binding and transport proteins following TBI. In the first phase, ferritin heavy chain and transferrin were increased at 7 days post-irradiation and fell to near basal levels thereafter. Ferritin heavy chain is the primary storage complex for Fe^3+^ iron within macrophages while transferrin is the primary protein for the secure transport of Fe^3+^ through the plasma [23]. The increased expression of these proteins at 7 days post-irradiation suggests an early transient phase of increased iron storage in ferritin. In the second phase of regulation, we observed delayed upregulation of CD71/transferrin receptor, integrin alphaM/Mac-1, and lipocalin-2. CD71/transferrin receptor was elevated transiently from 14–21 days post-irradiation. Increased CD71/transferrin receptor was expressed on early-stage RBC, but also on cells that were histologically similar to mature RBC. The expression of CD71/transferrin receptor on late-stage RBC in the spleen was previously demonstrated to occur in response to erythropoietic stress [42]. Integrin alphaM and lipocalin-2 were elevated from 21–28 days post-irradiation. CD71/transferrin receptor binds transferrin-bound iron for import into the cell, while integrin alphaM has been shown to bind to and import iron oxide nanoparticles [33,34]. Lipocalin-2 participates in the regulation of iron homeostasis [36,37] and safely sequesters and transports iron [35,36]. These data suggest a secondary phase of uptake of transferrin, regulation of iron homeostasis, and possibly uptake of iron particles. Finally, in the third pattern of expression, we observed suppression of *Flvcr1* gene expression and CD163 over the entire time course of the experiment. FLVCR1a functions to export heme-bound iron from macrophages and other cell types [38]. CD163 is a high-affinity scavenger receptor for the hemoglobin-haptoglobin complex [54]. Importantly, although we observed an increase in ferroportin gene expression, we did not observe any significant change in ferroportin protein, potentially due to increased hepcidin expression from the liver. Hepcidin suppresses iron export in reticulendothelial macrophages and duodenal enterocytes [28,40]. Suppression of ferroportin would result in iron retention in the spleen. Together, these data suggest that following radiation-induced hemolysis, there is a suppression of proteins that transport heme iron complexes but biphasic upregulation of other iron binding and transport proteins. The mechanisms that regulate the timing of these events following radiation are not known.

The presence of high levels of iron in the spleen correlates with the activation of programmed cell death, as indicated by caspase-3 activation. We also identified the alteration of a number of markers of ferroptosis, including upregulation of transferrin receptor and COX-2, downregulation of GPX-4 and SLC7A11, increased tissue iron levels, and increased protein-linked MDA. The significance of increased transferrin receptor expression is that this receptor is believed to provide transport of iron into the cell, where it may then participate in the generation of reactive oxygen species (ROS) if it is not efficiently sequestered. The GPX-4/SLC7A11 proteins provide a mechanism of protection against iron-induced ROS by detoxification; the downregulation of these proteins is consistent with the lack of antioxidant protective response within the cell. COX-2 protein has been shown to be a marker, but not a driver, of ferroptosis. Further investigation, to determine the presence of ROS, lipid peroxidation, and mitochondrial status, is needed to confirm the activation of ferroptosis and to determine the cell types that undergo programmed cell death in the spleen following TBI.

Iron and radiation have both been shown to modulate macrophage polarity [23,38,50]. Our data suggest that TBI would necessarily result in both radiation damage to macrophages and their exposure to high levels of iron. Our cell culture murine macrophage studies indicate that iron alone does not significantly upregulate the pro-inflammatory M1 polarization markers tested, whereas radiation exposure and iron + radiation can upregulate iNOS and IL-1β. Arg-1, an anti-inflammatory M2 polarization marker, can be upregulated by combined radiation + iron. Neither iron, radiation, nor the combination inhibited cytokine-induced M1 or M2 polarization. While our investigation of macrophage polarization is not exhaustive, our data suggest that combined iron + radiation may induce a mixed polarity. This issue also merits further exploration, especially given the importance of delayed inflammation following radiation exposure.

Our previous studies showed that captopril mitigates H-ARS in mice and Göttingen minipigs [28,30]. Captopril treatment without radiation does induce ~10% reduction in hematocrit, associated with suppression of erythropoietin expression [55,56]. However, although captopril and other ACE inhibitors have been used in medicine for the treatment of hypertension and heart failure for over forty years [57], hemolysis has not been found to be an adverse effect of ACE inhibitors [57]. In fact, several ACE inhibitors have been shown by Hayase et al. (2003) to not induce hemolysis [58]. Our current data indicate that captopril treatment did not mitigate the radiation-induced reduction of RBC or HGB, or deposition of iron in the spleen. Consistent with this, captopril did not modulate most of the iron binding or iron storage proteins. Captopril treatment after irradiation enhanced integrin alphaM and lipocalin-2 gene expression. As integrin alphaM is also a macrophage marker, this could indicate improved recovery of macrophages or improved iron homeostasis. The spleen is a significant location for extramedullary hematopoiesis in mice, and the increased weight of the spleen (around day 21 post-irradiation) may be related to hematopoietic recovery [59,60]. Increased understanding of the changes in cell populations in the spleen following TBI requires flow cytometry analysis, which we hope to perform in future studies.

In summary, our findings indicate that the spleen is a major organ for the deposition of iron following exposure to ionizing radiation. Iron deposition results in the regulation of a variety of iron binding and transport proteins and correlates with programmed cell death in the spleen. Additionally, iron and radiation together may affect the polarization of tissue macrophages, potentially inducing mixed polarity. Improved understanding of the effects of iron as a secondary toxicity following TBI may allow for advancements in the treatment of delayed radiation tissue damage.

## 4. Methods

### 4.1. Chemicals

Reagents were obtained from Millipore Sigma (St. Louis, MO, USA) except where indicated.

### 4.2. Animals, Captopril Treatment, Irradiation and Tissue Collection

Female C57BL/6J mice were purchased from Jackson Laboratories (Bar Harbor, ME, USA). Mice were kept in a barrier facility for animals accredited by the Association for Assessment and Accreditation of Laboratory Animal Care International. Mice were housed in groups of four. Animal rooms were maintained at 21 ± 2 °C, 50% ± 10% humidity, and 12 h light/dark cycle with freely available rodent ration (Harlan Teklad Rodent Diet 8604, Frederick, MD, USA). Animals were randomized to treatment groups for each experiment, using the appropriate numbers of animals to ensure significance. At 12–14 weeks of age, mice were placed in Lucite jigs and exposed to TBI in a bilateral gamma radiation field in the Armed Forces Radiobiology Research Institute (AFRRI, Bethesda, MD, USA) high-level ^60^Co facility as previously described [61]. The midline tissue dose for the mice was 6.85 Gy at a dose rate of 0.6 Gy/min, a sub-lethal radiation exposure for this strain of mice. The alanine/electron spin resonance (ESR) dosimetry system was used to measure dose rates (to water) in the cores of acrylic mouse phantoms [62]. Groups were sham irradiated + vehicle, irradiated + vehicle, and irradiated + captopril. Sham irradiated control groups were placed in Lucite jigs but without exposure to radiation; the sham irradiated animals were used to determine basal or control levels in experiments. Captopril (USP grade; Millipore Sigma) was dissolved at 0.13 g/L in acidified water [63] to deliver ~13–26 mg/kg/day [30,55]. Captopril in the water was provided to animals from 2 days post-irradiation for 14 consecutive days [30,55]. Control animals received acidified water (vehicle) without captopril. All animals in this study also received injections of saline (100 µL subcutaneously) on the same days as captopril administration, as these groups were control groups from another study. Euthanasia was performed by injection of 0.1–0.2 mL Fatal Plus (Vortech Pharmaceuticals, Dearborn, MI, USA; 39–78 mg pentobarbital) per animal in accordance with current American Veterinary Medical Association Guidelines for Euthanasia. Euthanasia was ensured prior to the collection of tissues. Blood was obtained by cardiocentesis as previously described [30]. Complete blood counts (CBC) with differentials to directly measure reticulocyte numbers were obtained using a Baker Advia 2120 Hematology Analyzer (Siemens, Tarrytown, NY, USA). Separate cohorts of mice were used at each time point. 

### 4.3. Histology and Immunohistochemistry (IHC)

Spleens were surgically removed from euthanized animals and fixed in 10% neutral buffered formalin overnight. Tissues were processed and embedded in paraffin using standard methods and stained for hematoxylin and eosin (H andE) and Prussian blue staining (Histoserv, Inc., Germantown, MD, USA). For immunohistochemistry (IHC), antigen retrieval was performed by heating slides for 15 min at 95 °C in 1 mM EDTA (pH 8). Sections were blocked by incubation with 4% normal goat serum (Vector, Burlingame, CA, USA) for 1 h at room temperature. Slides were incubated in primary antibody overnight at 4 °C at these concentrations: anti-ferritin (1:400, Millipore Sigma, St. Louis, MO, USA), anti-CD71 (1:200, Invitrogen, Waltham, MA, USA). Sections were then incubated in 1:1000 dilution goat anti-rabbit secondary antibody (Jackson Immunoresearch, West Grove, PA, USA) followed by incubation in ABC-AP (Vector) according to kit instructions. Immunoreactivity was visualized with ImmPACT Vector Red Substrate (Vector) per kit instructions and sections were dehydrated, cleared, and coverslipped with Permount mounting media (ThermoFisher, Waltham, MA, USA). Spleen sections were digitally scanned using the Ziess Axioscan for analysis and low magnification images for publication were produced with Zen Lite software (Carl Ziess Meditech, Inc., Dublin, CA, USA). ImageJ software was used for histological image analysis (NIH, Bethesda, MD, USA) [64]. Stained slides were evaluated by a pathologist who was blinded to the identity of the treatment groups. High magnification images were taken to obtain representative images from 3 separate slides, 30× magnification, using a Nikon Eclipse Ti microscope (Minato City, Tokyo, Japan).

### 4.4. Cell Culture, Irradiation, and Iron Treatment

The J774A.1 murine macrophage cells (American Tissue Culture Collection, Manasses, VA, USA) were cultured in Dulbecco’s Modified Eagles Medium (Quality Biological, Gaithersburg, MD, USA; Catalog #:112-300-101), 10% fetal bovine serum (Gemini Bio, West Sacremento, CA, USA), with penicillin (100 units/mL) and streptomycin (100 µg/mL), 0.5% fungizone (Gibco/ThermoFisher Scientific, Waltham, MA, USA). Primary human spleen microvascular endothelial cells (HSMVEC; Cell Biologics, Chicago, IL, USA) were grown in a microvascular endothelial complete medium with the gelatin-based coating solution, according to the manufacturer’s instructions. Passages 3–5 were used for experiments. Cells were maintained in a humidified environment of 5% CO_2_/95% air at 37 °C. One day prior to irradiation, J774A.1 cells were plated at 7.6 × 10^4^ cells/mL in 12 well plates (CellTreat Scientific Products, LLC, Pepperell, MA, USA). For HSMVEC, cells were cultured to 70–90% confluence in 30 mm tissue culture plates. Cells were irradiated using an RS2000 Biological Irradiator (Rad Source Technologies, Alpharetta, GA, USA) at a dose rate of 1.15 Gy/min (160 kV, 25 mA) for a total dose of 2 or 6.8 Gy as previously described [53,65]. For exposure to iron, cells were treated with a filter sterilized solution of ammonium iron III citrate (Merck, Inc., Billerica, MA, USA) and added to a final concentration of 7 or 12 mg/L in cell culture medium. For the treatment of cells with radiation and iron, irradiation was performed prior to the addition of iron.

### 4.5. RNA Purification, Reverse Transcription Polymerase Chain Reaction (RT-PCR)

Spleen tissues harvested after euthanasia were immediately placed in RNALater buffer and stored according to the manufacturer’s instructions (Qiagen, Germantown, MD, USA). At the time of RNA purification, tissues were homogenized with an Ultra Turrax homogenizer (Jahnke & Kunkel, Staufen, Germany). Tissue samples were then processed using QIAshredder mini columns (Qiagen). RNA was isolated from the tissue homogenate using the RNeasy mini kit (Qiagen), and genomic DNA was removed using the RNase-free DNase Set (Qiagen). For J774A.1 or HSMVEC, media was removed, and the cells were washed twice with cold PBS. Cells were lysed in RNAlater (Qiagen, Rockville, MD, USA) with β-mercaptoethanol. Total RNA was isolated from cultured cells using the RNeasy Mini Kit (Qiagen, Valencia, CA, USA) according to the manufacturer’s protocol. Purified RNA, from spleen tissue or cultured cells, was quantified spectroscopically (ND-1000 Spectrophotometer; NanoDrop Technologies, Wilmington, DE, USA). 500 ng of RNA was reverse transcribed in 20 µl using the iScript cDNA synthesis kit (Bio-Rad, Hercules, CA, USA), according to the manufacturer’s protocol. Reverse transcription of cDNA was performed using an iCycler with an IQ5 optical system (Bio-Rad, Hercules, CA, USA). cDNA was then diluted with nuclease-free water to 2 ng/µL. qPCR was performed using four biological replicates for the captopril group and three for the radiation vehicle group for spleen samples. Three biological replicates were used for cell culture samples. 5 or 10 ng total complementary DNA (cDNA) was used in each 20 µL RT-qPCR reaction. RT-qPCRs were performed with technical duplicates using 12 µM of each primer and 10 µL of iTaq™ Universal SYBR^®^ Green Supermix (Bio-Rad), on a CFX96 Touch Real-Time PCR Detection System (Bio-Rad). Primers for qRT-PCR were designed using NCBI/Primer-BLAST and purchased from Integrated DNA Technologies (Coralville, IA, USA). Sequences for qPCR primers for *Mus musculus* are shown in Table 1. The qPCR primers for *Homo sapiens* are shown in Table 2. Relative gene expression to the housekeeping genes was calculated using the ΔΔCq method using CFX Maestro software, 2.0 (Bio-Rad) [66,67]. For quantification, the comparative threshold cycle (CT) method was performed using Microsoft Excel 13 (Microsoft, Redmond, WA, USA) to assess relative changes in mRNA levels between the untreated control and the drug- or iron-treated and/or irradiated samples.

### 4.6. Western Blotting

Western blots were performed as described [68]. Cells or tissues were incubated in RIPA buffer for 20 min at 4 °C, and then centrifuged at 10,000 RPM for 7 min 4 °C. Cells were lysed in RIPA buffer (1% NP-40, 0.1% SDS, 0.1% Na-deoxycholate, 10% glycerol, 0.137 M NaCl, 20 mM Tris pH [8.0]) (ThermoFisher Scientific, Waltham, MA, USA), with protease inhibitors (#A32953, ThermoFisher) and phosphatase inhibitors (#A32957, ThermoFisher). Proteins were resolved by SDS-PAGE (Criterion TGX precast, Bio-Rad) and transferred to a nitrocellulose membrane. Western blot protein bands were detected and quantified using the Odyssey system (LI-COR). Proteins were detected by blotting using anti-ferritin (Santa Cruz Biotechnology, Inc., Santa Cruz, CA, USA; #sc-256; 1:1000), anti-CD71 (Santa Cruz, #sc-258; 1:500), anti-β-actin (Millipore Sigma #AC-15; 1:5000), anti-p21/waf1 (Cell Signaling Technologies, Danvers, MA, USA; #64016; 1:1000), anti-cleaved caspase-3 (Cell Signaling #9662; 1:1000), anti-malondialdehyde (MDA) for detection of modified proteins (Alpha Diagnostic International, San Antonio, TX, USA, #MDA11-S: 1:1000), and anti-Cox-2 (Cell Signaling #12282; 1:1000) primary antibodies. Note that for anti-malondialdehyde detection of modified proteins, samples were run in the absence of a reducing agent. Antibodies were diluted in the buffers recommended by the manufacturers and incubated at 4 °C for 1–5 days with gentle rocking. Anti-mouse and anti-rabbit secondary antibodies conjugated to IRDye680 or IRDye800 (LI-COR, Lincoln, NE, USA; 1:10,000) were used to probe primary antibodies, according to the manufacturer’s instructions. Proteins were normalized to β-actin which was used to probe the same gel. Gel images were prepared using Odyssey System software and Corel Draw X7 Graphics software (Corel Corporation, Ottawa, ON, Canada), in accordance with the Author Guidelines. Full gels are provided in the Appendix A.

### 4.7. Iron Quantification 

Levels of total iron (II + III) were determined in spleen homogenates using the Iron Colorimetric Assay Kit (Biovision Inc., Milpitas, CA, USA) in accordance with the manufacturer’s instructions. After the reduction of total iron in the samples, ferrous ion was detected by monitoring the absorbance at 593 nm of the reaction with Ferene S using a CYTATION 5 Imaging Reader (BioTek Instruments, Winooski, VT, USA).

### 4.8. Statistical Analysis

Statistical analysis was performed using GraphPad Prism V6 (GraphPad Prism Software, Inc., San Diego, CA, USA). Results are represented as means ± SEM. *p* values of <0.05 were considered significant. Two-way ANOVA with either Tukey or Sidak posthoc tests was used for multiple comparisons. 

## Figures and Tables

**Figure 1 ijms-23-11029-f001:**
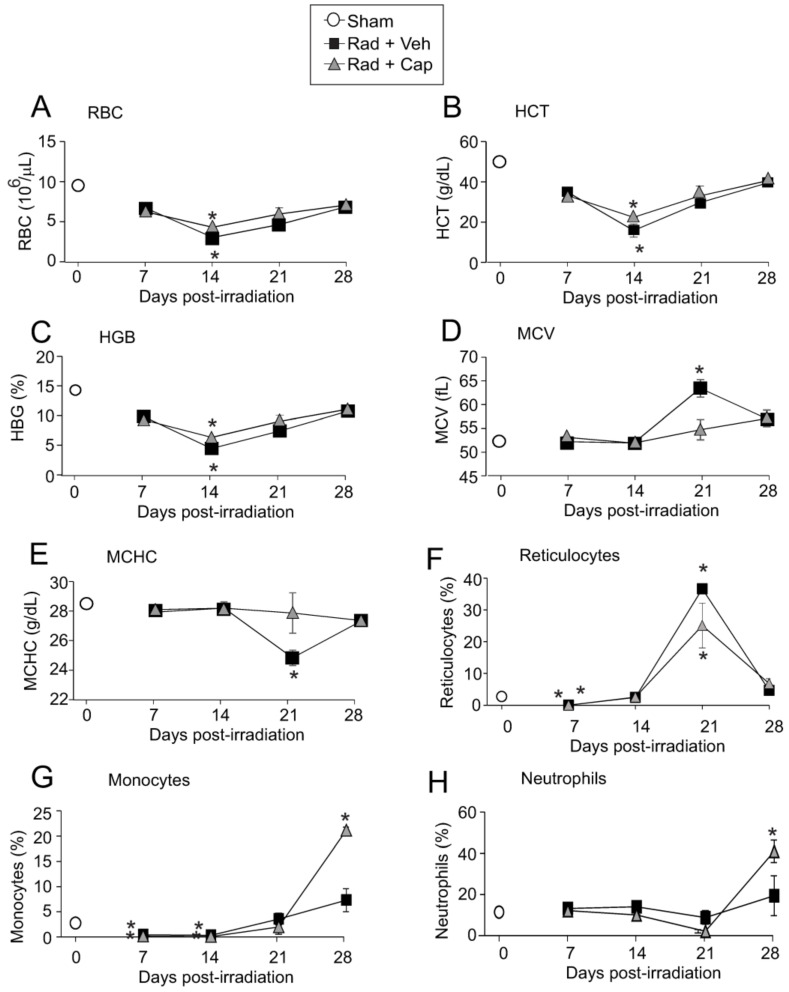
**Total body irradiation causes decreased red blood cell, hematocrit, and hemoglobin levels.** C57BL/6 mice were exposed to 6.85 Gy total body irradiation. Animals received vehicle (drinking water alone) or captopril in the drinking water from days +2–+16 post-irradiation. At the indicated time points, mice were euthanized, and blood was obtained for complete blood cell counts. Control (Sham) blood cell levels are indicated by an open circle at day 0. (**A**) red blood cells (RBC); (**B**) hematocrit (HCT); (**C**) hemoglobin (HGB); (**D**) mean corpuscular volume (MCV); (**E**) mean corpuscular hematocrit (MCHC); (**F**) reticulocytes; (**G**) monocytes; and (**H**) neutrophils. Data show means ± SEM, from *n* = 3 animals per group; * indicates *p* < 0.05 from control (sham irradiated) levels.

**Figure 2 ijms-23-11029-f002:**
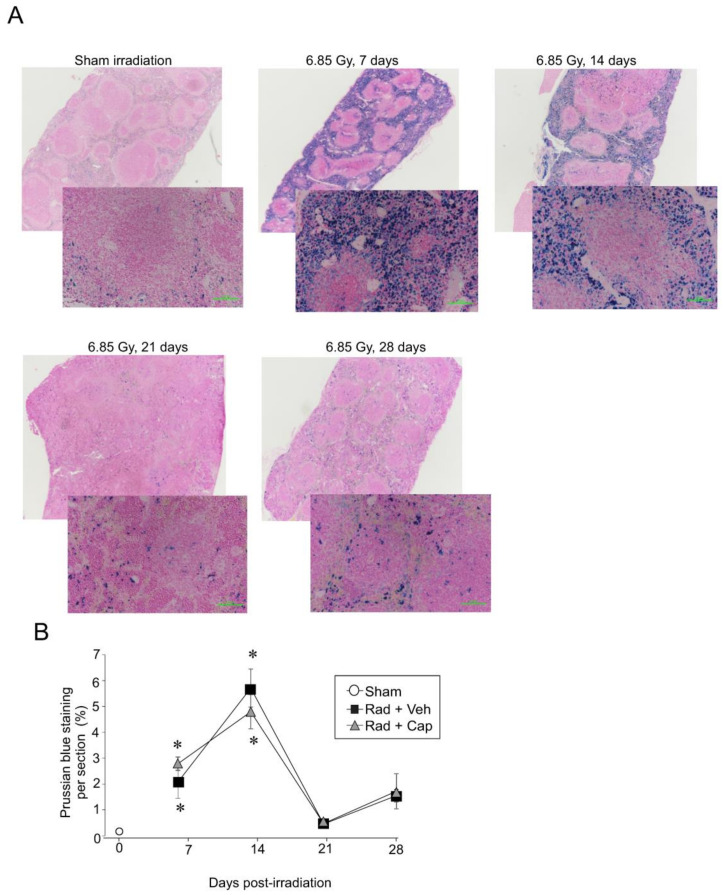
**Total body irradiation is associated with increased Fe^3+^ deposition in the spleen.** C57BL/6 mice were exposed to 6.85 Gy total body irradiation. Animals received vehicle (drinking water alone) or captopril in the drinking water on days +2–+16 post-irradiation. At the indicated time points, mice were euthanized, and spleen tissue was obtained for histology. (**A**) Spleen sections were stained with Prussian blue to visualize Fe^3+^ content from the animal receiving vehicle. Whole spleen images are shown from the Ziess Axioscan. Representative images from 3 separate slides are shown in the high-powered (30×) magnification. (**B**) Prussian blue staining was quantified. Graph shows means ± SEM from *n* = 4 animals per group; * indicates *p* < 0.05 from control (sham irradiated) levels.

**Figure 3 ijms-23-11029-f003:**
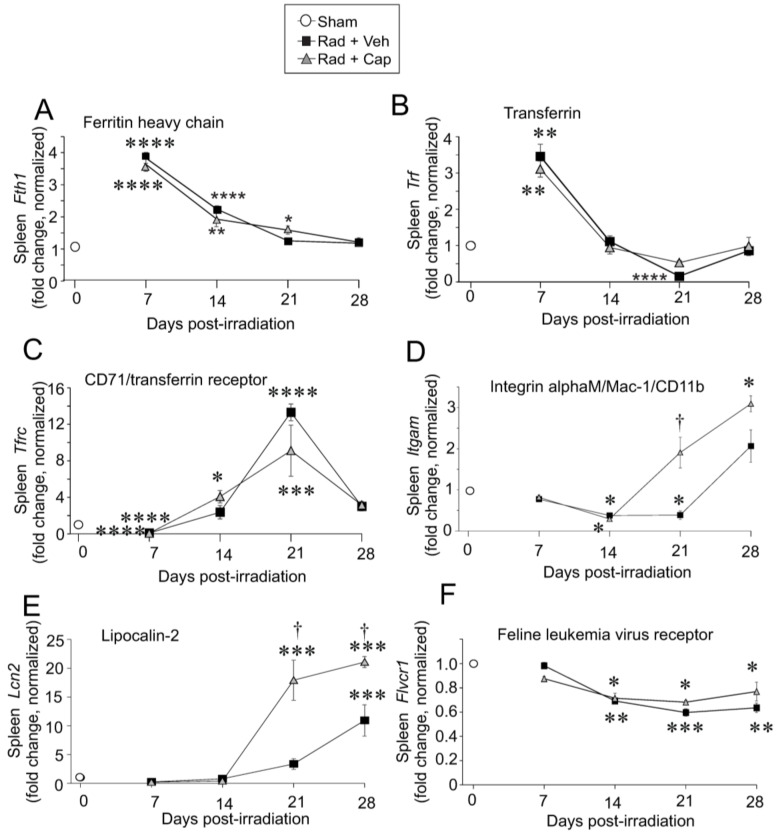
**Total body irradiation results in increased gene expression for iron handling proteins in the spleen.** C57BL/6 mice were exposed to 6.85 Gy total body irradiation. Animals received vehicle (drinking water alone) or captopril in the drinking water on days +2–+16 post-irradiation. At the indicated time points, mice were euthanized, and spleen tissue was obtained for RNA analysis. qPCR was performed for the following genes: (**A**) *Fth1* (ferritin heavy chain); (**B**) *Trf* (transferrin); (**C**) *Tfrc* (CD71/transferrin receptor1); (**D**) *Itgam* (integrin alphaM/Mac-1); (**E**) *Lcn2* (lipocalin-2); (**F**) *Flvcr1* (feline leukemia virus receptor). Graphs show means ± SEM from *n* = 4 animals per group. Asterisks: * indicates *p* < 0.05 from control (sham irradiated) levels; ** indicates *p* < 0.01 from control; *** indicates *p* < 0.001 from control; and **** indicates *p* < 0.0001 from control. Dagger (†) indicates that the captopril-treated group value is *p* < 0.05 from the vehicle-treated group at the same time point. Control (sham) levels of each gene are shown as open circles at time 0.

**Figure 4 ijms-23-11029-f004:**
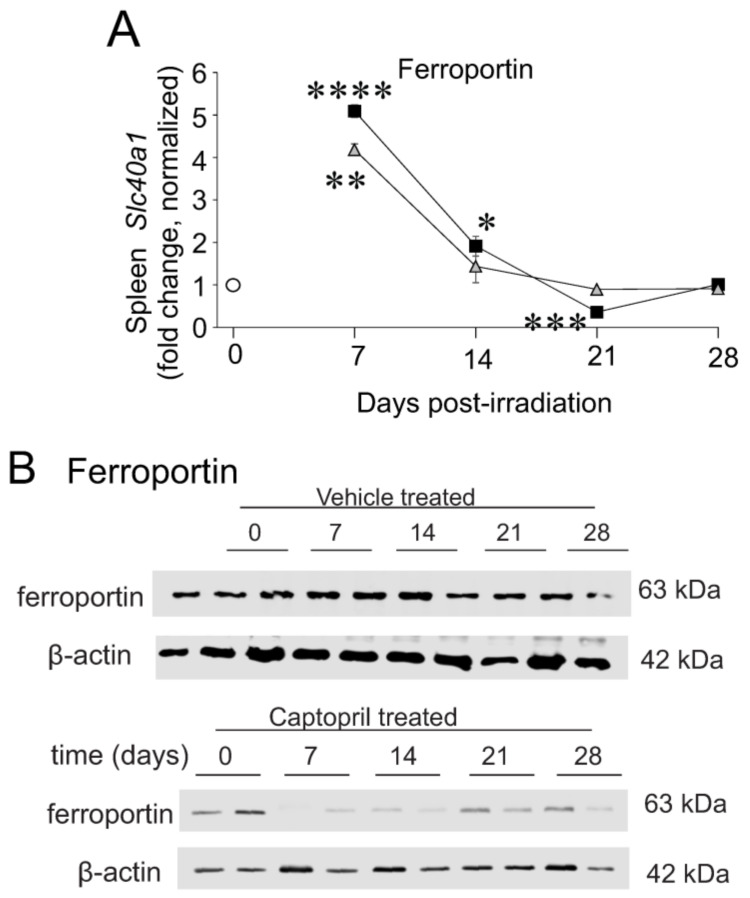
**Total body irradiation results in increased gene expression of ferroportin but not ferroportin protein.** C57BL/6 mice were exposed to 6.85 Gy total body irradiation. Animals received vehicle (drinking water alone) or captopril in the drinking water on days +2–+16 post-irradiation. At the indicated time points, mice were euthanized, and spleen tissue was obtained for RNA and protein analysis. (**A**) qPCR was performed *Slc40a1* (ferroportin). Graphs show means ± SEM from *n* = 4 animals per group. Asterisks: * indicates *p* < 0.05 from control (sham irradiated) levels; ** indicates *p* < 0.01 from control; *** indicates *p* < 0.001 from control; and **** indicates *p* < 0.0001 from control. (**B**) Western blotting of ferroportin. Blots were probed for β-actin as a loading control. Representative data are shown.

**Figure 5 ijms-23-11029-f005:**
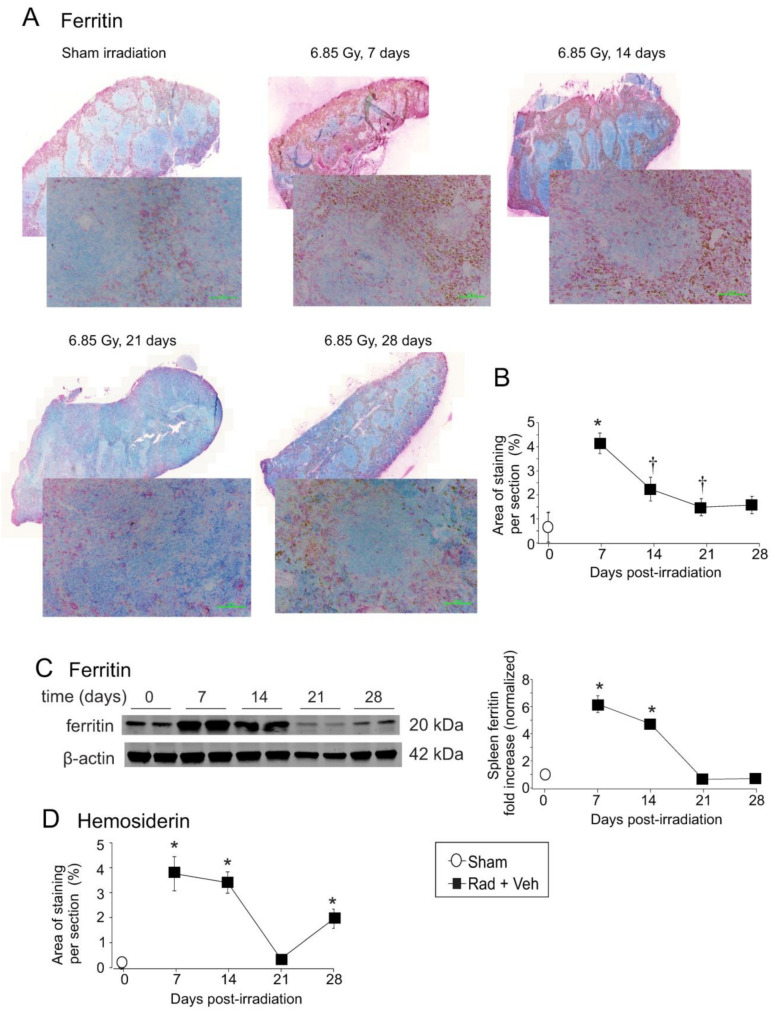
**Total body irradiation results in increased ferritin and hemosiderin in the spleen.** C57BL/6 mice were exposed to 6.85 Gy total body irradiation. At the indicated time points, mice were euthanized, and spleen tissue was obtained for immunohistochemistry and western blotting. (**A**) Spleen sections were stained for ferritin. Whole spleen images are shown from the Ziess Axioscan. Representative images from 3 separate slides are shown in the high-powered (30×) magnification. (**B**) Quantification of ferritin IHC staining. Graph shows means ± SEM, N = 3; † indicates *p* < 0.05 compared with 7 days post-irradiation. (**C**) Western blots of ferritin in total spleen tissue; blots were reprobed for β-actin as a loading control. Representative data are shown from *n* = 2 animals. Graph shows means ± SEM, *n* = 4 animals; * indicates *p* < 0.05 from control. (**D**) Quantification of hemosiderin staining in the spleen. Graph shows means ± SEM, *n* = 3; * indicates *p* < 0.05 compared with control.

**Figure 6 ijms-23-11029-f006:**
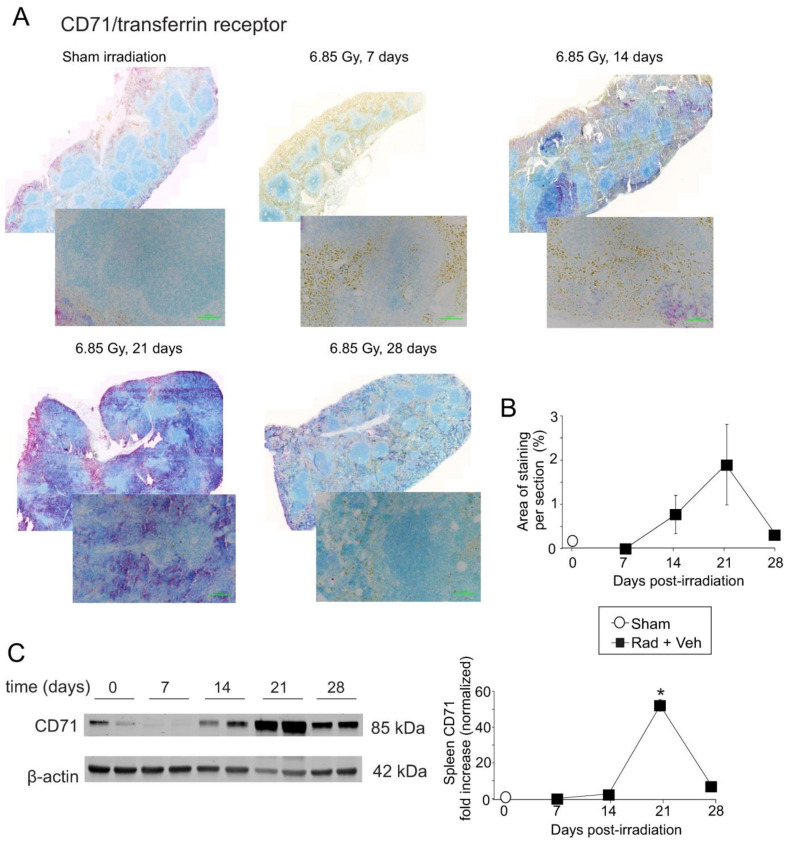
**Total body irradiation results in increased transferrin receptor/CD71 in the spleen.** C57BL/6 mice were exposed to 6.85 Gy total body irradiation. At the indicated time points, mice were euthanized, and spleen tissue was obtained for immunohistochemistry and western blotting. (**A**) Spleen sections were stained for transferrin receptor/CD71. Whole spleen images are shown from the Ziess Axioscan. Representative images from 3 separate slides are shown in the high-powered (30×) magnification. (**B**) Quantification of transferrin receptor/CD71 IHC staining. Graph shows means ± SEM, *n* = 3. (**C**) Western blots of transferrin receptor/CD71 in total spleen tissue; blots were reprobed for β-actin as a loading control. Representative data are shown from *n* = 2 animals. Graph shows normalized means ± SEM, *n* = 4 animals; * indicates *p* < 0.05 from control.

**Figure 7 ijms-23-11029-f007:**
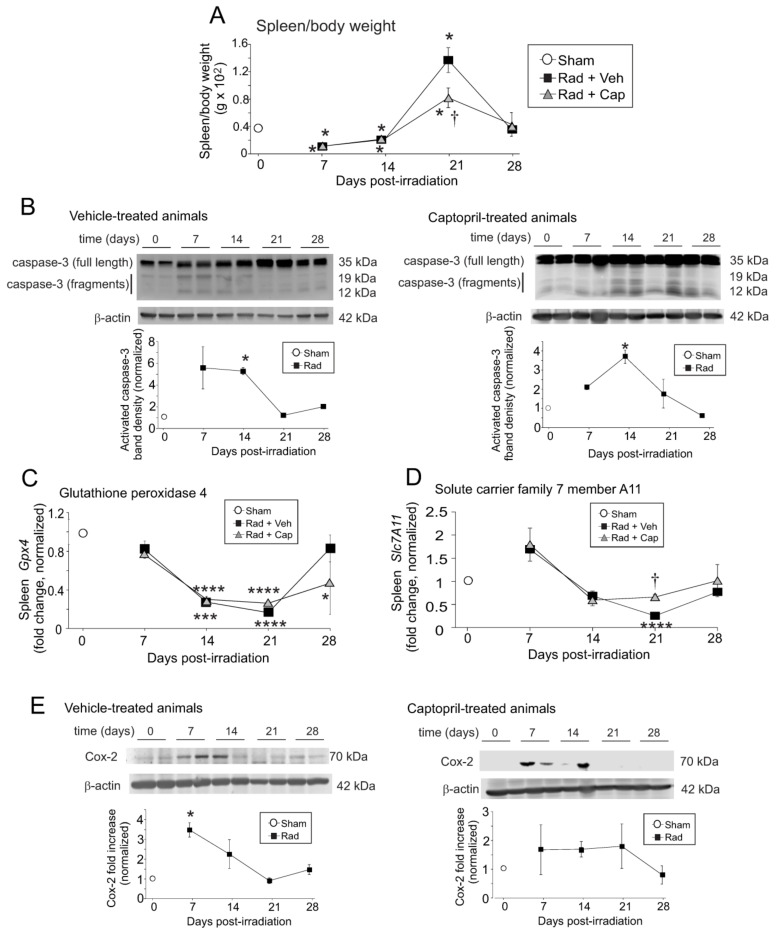
**Total body irradiation results in biphasic alterations in spleen weight and increased expression of markers for cell death and ferroptosis in the spleen.** C57BL/6 mice were exposed to 6.85 Gy total body irradiation. Animals received vehicle (drinking water alone) or captopril in the drinking water on days +2–+16 post-irradiation. At the indicated time points, mice were euthanized, and spleen tissue was obtained. Spleen weight and total body weights were obtained, and spleen tissues were used for western blotting and qPCR. (**A**) Spleen weights as a percentage of total body weight. Graph shows means ± SEM, n = 4 animals. * indicates *p* < 0.05 from control (0 time point). † indicates *p* < 0.05 compared with the vehicle-treated group at 21 days post-irradiation. (**B**) Western blots of total and activated caspase 3; blots were reprobed for β-actin as a loading control. The left panel shows vehicle-treated animals; the right panel shows captopril-treated animals. Representative data are shown from n = 2 animals from each group. Graphs show means ± SEM, n = 4 animals; * indicates *p* < 0.05 from control (0 time point). (**C**,**D**) Spleen tissues from sham irradiated (0 time point), vehicle-treated, or captopril-treated animals were used for qPCR for glutathione peroxidase 4 (*Gpx4*; (**C**)) or solute carrier family 7 A 11 (*Slc7a11*; (**D**)). Graphs show means ± SEM, n = 4 animals; * indicates *p* < 0.05, *** *p* < 0.001, and **** *p* < 0.0001 from control (0 time point). † indicates *p* < 0.05 from the vehicle-treated group at the same time point. (**E**) Western blots of COX-2; blots were reprobed for β-actin as a loading control. The left panel shows vehicle-treated animals; the right panel shows captopril-treated animals. Representative data are shown from *n* = 2 animals from each group. Graphs show means ± SEM, n = 4 animals; * indicates *p* < 0.05 from control (0 time point). (**F**) Iron assay of vehicle-treated spleen samples. Graphs show means ± SEM, *n* = 3–4 animals; * indicates *p* < 0.05 from control (0 time point). (**G**) Western blots of MDA-linked proteins in a non-reducing gel. Samples were probed for β-actin as a loading control in a denaturing gel. Representative data are shown. Graphs show means ± SEM, *n* = 3–4 animals; * indicates *p* < 0.05 from control (0 time point).

**Figure 8 ijms-23-11029-f008:**
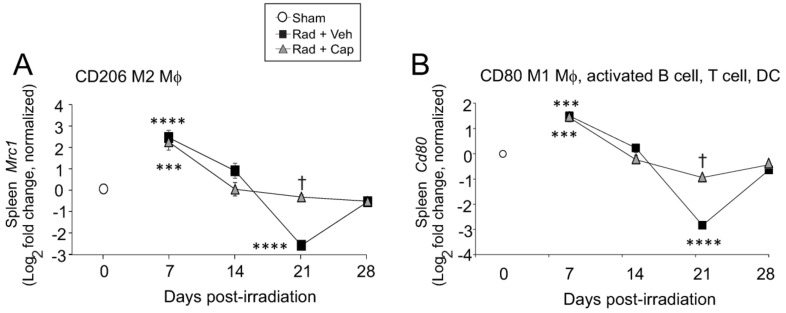
**Total body irradiation results in altered gene expression for macrophage polarity markers in the spleen.** C57BL/6 mice were exposed to 6.85 Gy total body irradiation. Animals received vehicle (drinking water alone) or captopril in the drinking water on days +2–+16 post-irradiation. At the indicated time points, mice were euthanized, and spleen tissue was obtained for RNA analysis. qPCR was performed for the following genes: (**A**) *CD206* (M2 macrophage [MΦ]); (**B**) *Cd80* (M1 macrophage [MΦ], activated B cell, T cell and dendritic cell [DC]). Graphs show means ± SEM from *n* = 4 animals per group. Asterisks: *** indicates *p* < 0.001 from control (sham); and **** indicates *p* < 0.0001 from control. † indicates *p* < 0.05 in the captopril-treated group from the vehicle-treated group at the same time point.

**Figure 9 ijms-23-11029-f009:**
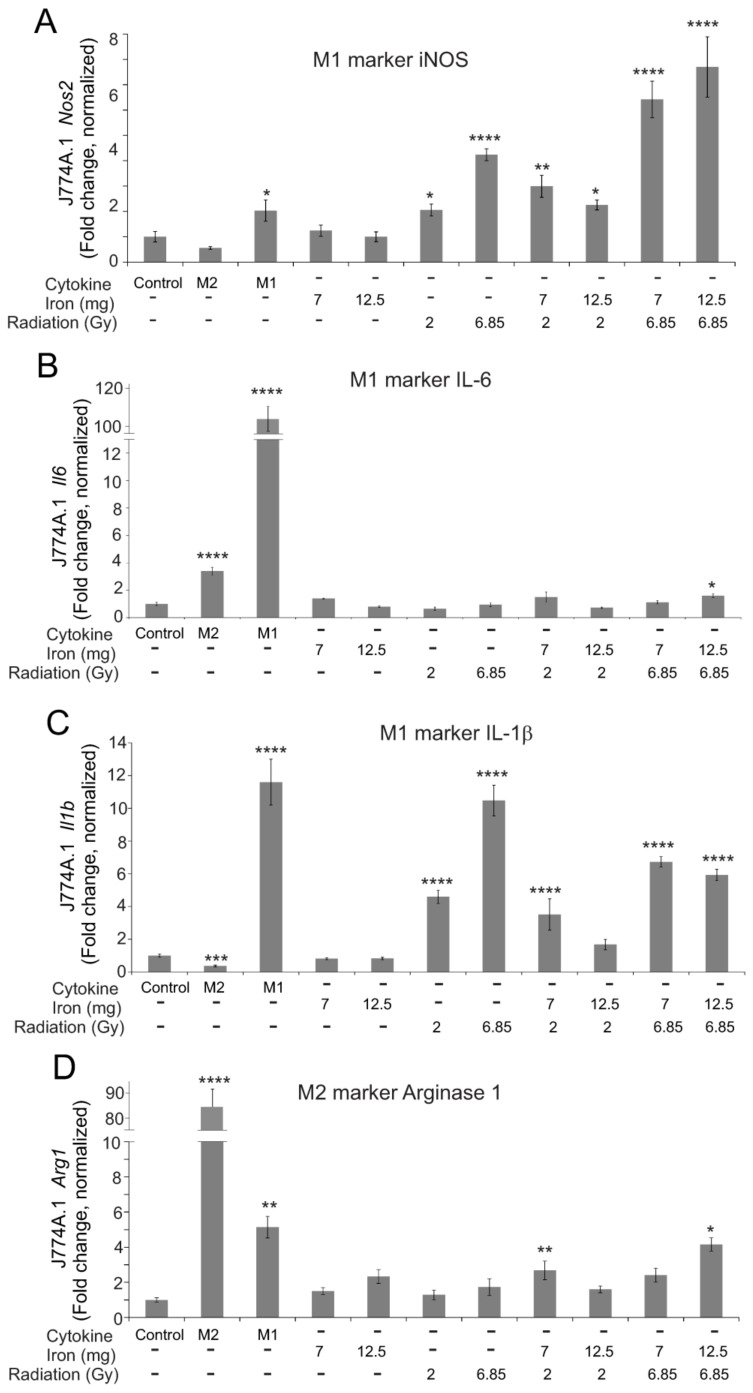
**Effects of radiation and high iron concentrations on J774A.1 murine macrophage polarity in culture.** J774A.1 cells were grown to ~80% confluence, and untreated (control) exposed to either 7 or 12 mg/L Fe^3+^ and/or 2 or 6.85 Gy X-ray irradiation. As a control for M2 polarization, cells were treated with 100 ng/mL IL-4; as a control for M1 polarization, cells were treated with 100 ng/mL IFN-γ + 100 ng/mL LPS. 24 h post-irradiation, RNA was purified. qPCR was performed for M1 markers (*Nos2* (**A**), *Il6* (**B**), and *Il1b* (**C**)), and an M2 marker (*Arg1* (**D**)). Gene expression was normalized to *Gapdh,* and fold increase was calculated relative to untreated cells (control). Graphs show means ± SEM, *n* = 3 independent experiments. Asterisks: * indicates *p* < 0.05 from control levels; ** indicates *p* < 0.01 from control; *** indicates *p* < 0.001 from control; and **** indicates *p* < 0.0001 from control.

**Figure 10 ijms-23-11029-f010:**
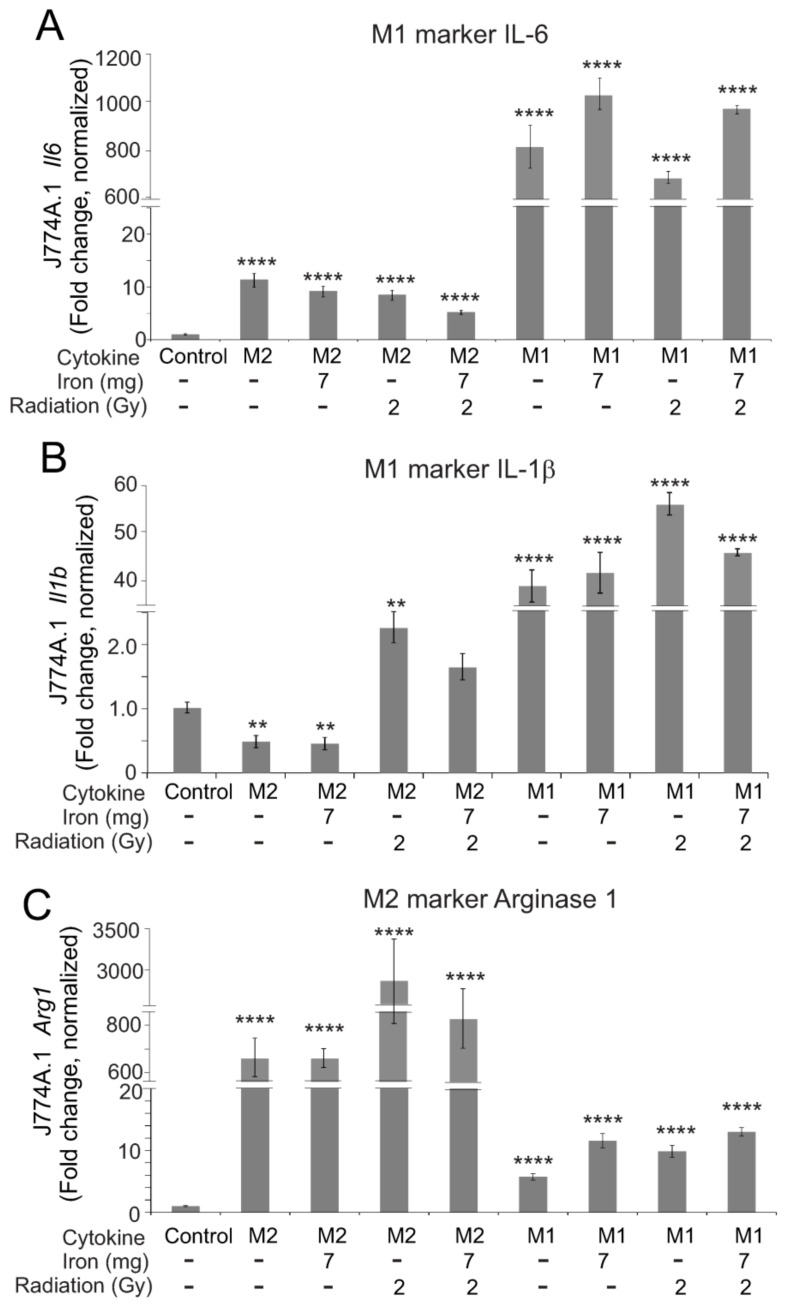
**Effects of radiation and high iron concentrations on J774A.1 murine macrophage polarization by cytokines in culture.** J774A.1 cells were grown to ~80% confluence, and untreated (control) exposed to either 7 or 12 mg/L Fe^3+^ and/or 2 or 6.85 Gy X-ray irradiation. 1 h after treatments, cells were treated with 100 ng/mL IL-4 to induce M2 or 100 ng/mL IFN-γ + 100 ng/mL LPS to induce M1. As a control for M2 and M1 polarization, cells were treated with IL-4 or IFN-γ + LPS with no pretreatment. 24 h post-irradiation, RNA was purified. qPCR was performed for M1 markers (*Il6* (**A**) and *Il1b* (**B**)), and an M2 marker (*Arg1* (**C**)). Gene expression was normalized to *Gapdh,* and fold increase was calculated relative to untreated cells (control). Graphs show means ± SEM, *n* = 3 independent experiments. Asterisks: ** indicates *p* < 0.01 from control; and **** indicates *p* < 0.0001 from control.

**Figure 11 ijms-23-11029-f011:**
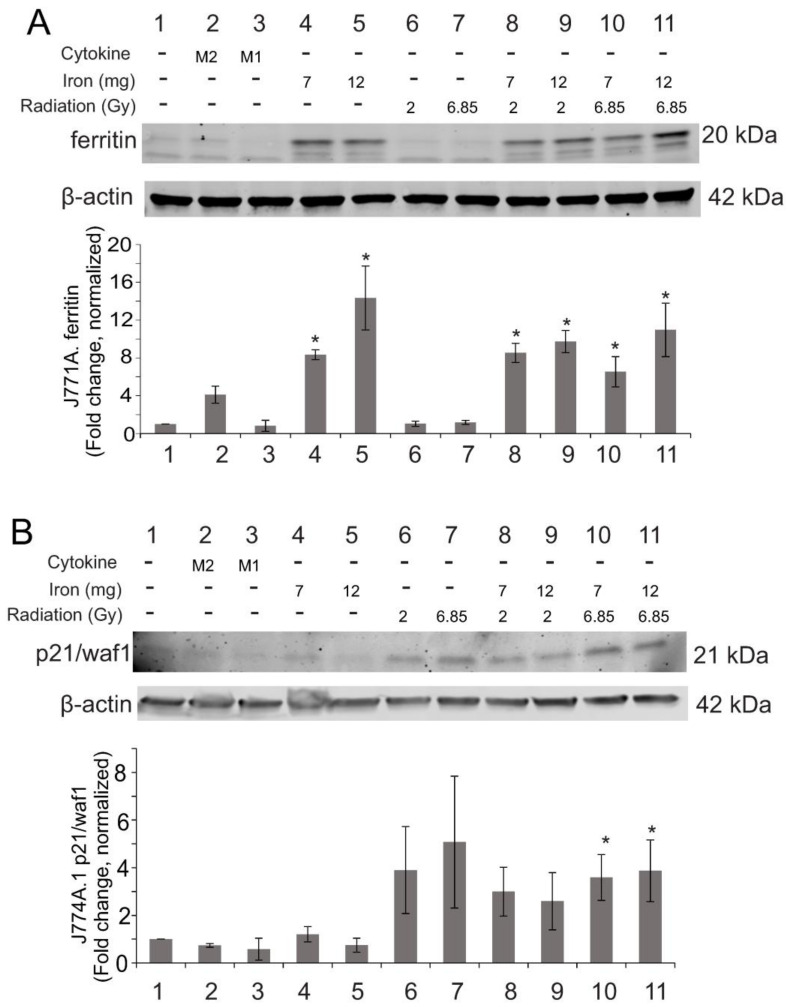
**Effects of radiation and high iron concentrations on J774A.1 murine macrophage expression of ferritin and senescence in culture.** J774A.1 cells were grown to ~80% confluence, and untreated (control) exposed to either 7 or 12 mg/L Fe^3+^ and/or 2 or 6.85 Gy X-ray irradiation. As a control for M2 polarization, cells were treated with 100 ng/mL IL-4; as a control for M1 polarization, cells were treated with 100 ng/mL IFN-γ + 100 ng/mL LPS. 24 h post-irradiation, Protein was purified and used for western blots for ferritin (**A**) or p21/waf1 (**B**). Blots were probed for β-actin as a loading control. Protein levels were normalized to β-actin and then the relative expression was determined compared with control levels. Graphs show means ± SEM, *n* = 3 independent experiments. * indicates *p* < 0.05 compared with control.

**Table 1 ijms-23-11029-t001:** Murine gene primers for qPCR.

Gene	Forward Primer	Reverse Primer
*Arg1*	5′-GGAACTCAACGGGAGGGTAAC-3′	5′-TAGTCCTGTCTGCTTTGCTGTGAT-3′
*Cd80*	5′-AGTTTCTCTTTTTCAGGTTGTGAA-3′	5′-ACATGATGGGGAAAGCCAGG-3′
*Cd206*	5′-CCACAGCATTGAGGAGTTTG-3′	5′-ACAGCTCATCATTTGGCTCA-3′
*Flvcr1*	5′-GGCACAATATAAACACCGGGC-3′	5′-TCCGACTGTATAGACACCATGAC-3′
*Fth1*	5′-AGTGCGCCAGAACTACCAC-3′	5′-AGCCACATCATCTCGGTCAA-3′
*Gapdh*	5′-ATGTGTCCGTTGTGGACTTG-3′	5′-GGTCCTCAGTGTAGCCCAAG-3′
*Gpx4*	5′-CGCCAAAGTCCTAGGAAACG-3′	5′-AAGGTTCAGGAATGGGCTCC-3′
*IL1b*	5′-AGTTGACGGACCCCAAAAG-3′	5′-AGCTGGATGCTCTCATCAGG-3′
*Il6*	5′-CCCCAATTTCCAATGCTCTCC-3′	5′-CGCACTAGGTTTGCCGAGTA-3′
*Itgam*	5′-AGAACACCAAGGACCGTCTG-3′	5′-AATCCAAAGACCTGGGTGCG-3′
*Lcn2*	5′-GGACTACAACCAGTTCGCCA-3′	5′-CAAAGCGGGTGAAACGTTCC-3′
*Nos2*	5′-CTTTTTCCCGGAGATGGGGG-3′	5′-GAGCTTGGCTTGGTACAGTCT-3′
*Slc11a11*	5′-TCCGAGGAGCAAGAGGAGTAA-3′	5′-TCCCCTTTGCTATCACCGAC-3′
*Slc40a1*	5′-TTCCTCCTCTACCTTGGCCA-3′	5′-CTGCCACCACCAGTCCATAG-3′
*Trf1*	5′-AAGTGCATCAGCTTCCGTGA-3′	5′-AGACCACACTGGCCTTGATG-3′
*Tfrc1*	5′-GCTCGTGGAGACTACTTCCG-3′	5′-AGAGAGGGCATTTGCGACTC-3′

Murine sequences for *Arg1* (arginase 1), *Cd80* (cluster of differentiation 80), *Cd206* (cluster of differentiation 206), *Flvcr1* (feline leukemia virus receptor; heme iron receptor), *Fth1* (ferritin heavy chain 1), *Gapdh* (glyceraldehyde-3-phosphate dehydrogenase), *Gpx4* (glutathione peroxidase 4), *Il1b* (interleukin 1 beta), *Il6* (interleukin-6), *Itgam* (integrin subunit alpha M), *Lcn2* (lipocalin-2; also known as neutrophil gelatinase associated lipocain or siderocalin), *Nos2* (inducible nitric oxide synthase), *Slc7a11* (solute carrier 7 member 11), *Scl40a1* (solute carrier 20 member 1, ferroportin), *Tfrc1* (transferrin receptor-1), *Trf1* (transferrin-1).

**Table 2 ijms-23-11029-t002:** Human gene primers for qPCR.

Gene	Forward Primer	Reverse Primer
*CDKN1A*	5′-GCCGAAGTCAGTTCCTTGTG-3′	5′-TCGAAGTTCCATCGCTCACG-3′
*FTH1*	5′-CCAGAACTACCACCAGGACTC-3′	5′-GAAGATTCGGCCACCTCGTT-3′
*GAPDH*	5′-AGCCACATCGCTCAGACAC-3′	5′-GCCCAATACGACCAAATCC-3′
*TFRC1*	5′-AGGACGCGCTAGTGTTCTTC-3′	5′-CCAGGCTGAACCGGGTATATG-3′

Human sequences for *CDKN1A* (p21/waf1), *FTH1* (ferritin heavy chain 1), *GAPDH* (glyceraldehyde-3-phosphate dehydrogenase), and *TFRC1* (transferrin receptor-1).

## Data Availability

All data is presented in the manuscript or is provided in the Supplementary Data Section. Any additional data is available upon request to the corresponding author.

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
