# Peer review of "Iron Deposition and Ferroptosis in the Spleen in a Murine Model of Acute Radiation Syndrome"

_ijms, 2022, doi:10.3390/ijms231911029_

Round 1
Reviewer 1 Report
The manuscript Iron Deposition and Ferroptosis in the Spleen in a Murine Model of Acute Radiation Syndrome investigates the effect of sub-lethal TBI on RBC hemolysis and subsequent splenic iron deposition.
The manuscript is very well written and the study is well designed with a robust analysis of the mechanisms underpinning the observed phenomena. This study investigates a radiation-induced effect that provides important data on non-DNA endpoints that may help to unravel mechanisms that contribute to secondary toxicity and longer-term radiation effects.
The experimental design was well thought out, using in vivo and in vitro models and several analytical techniques to provide a sound basis for the authors' discussion and conclusions. This study helps to emphasize the importance of in vivo models for investigating complex systemic responses to TBI.
Overall, the discussion is well organized and sufficiently cited to place the current investigation in the known literature. The authors' conclusions are reasonable and supported by their data. Future research plans are well thought out and will provide interesting insights into this investigation.
The use of a radiation countermeasure was an interesting addition to the study that provided valuable data regarding how the protection of bone marrow precursors impacts the hematopoietic response to TBI. This isn't a suggestion for the current study, but it would be interesting to see the effect of direct free radical scavaging would have on the same models.
There are no major issues with the manuscript, the minor issues are categorized below:
All figures and images- blurry and difficult to read, higher resolution images of all are required. Please show where the magnified square of the IHC image is located.
page 4
line 133: word missing after erythrocyte
page 5
line 168: Ferritin is the common protein for protein storage - underlined should be replaced with iron
page 18
lines 514 - 525: font size needs to be reduced
Author Response
We thank the reviewer for the comments. Below are our responses.
All figures and images- blurry and difficult to read, higher resolution images of all are required. Please show where the magnified square of the IHC image is located.
We will import higher resolution images for the revised manuscript. The magnified squares are representative areas, but they are not all taken from the same slide as the scanned full spleen images. We performed Prussian blue staining on 3 separate slides, and the magnified images were obtained with a different microscope, using one of the 3 slides for each condition. Therefore it is not possible to identify the area in squares since they were not all obtained from the same slide as the low powered image.
page 4
line 133: word missing after erythrocyte
This correction has been made.
page 5
line 168: Ferritin is the common protein for protein storage - underlined should be replaced with iron
This correction has been made.
page 18
lines 514 - 525: font size needs to be reduced
This correction has been made.
Reviewer 2 Report
This is a very interesting, valuable, competent and wide-ranging study of the effects of increasing iron concentration in the mouse system following quite a hefty, but sublethal, dose of external X-radiation. It is well designed and results are well described and discussed.
Author Response
We thank the reviewer for their comments.
Reviewer 3 Report
The manuscript by Rittase et al. presented that total body irradiation caused iron deposition and ferroptosis in the spleen by enhanced hemolysis of RBC.
Major concerns:
1. Please improve the resolution of all pictures and IHC slides. Most of the figures are difficult to present the differences.
1. In Figure 1, there was no sham group data which cohort paradelle to compare with radiation treatment.
2. In Figure 2A, was the IHC from radiation + vehicle or radiation + captopril? Please show both IHC staining results in Figure 2A.
3. Please label the protein name in the left western blots in figure 4B.
4. It is tough to identify the difference between IHC slides in Figure 5. Please improve the resolution of the slides.
5. In Figure 6A, IHC slides are not clear to check the ferritin staining. In Figure 6B, the error bar lines were not equal. Please improve them. Whether the combination of radiation and captopril cause a similar effect on ferritin compared to radiation alone?
6. In Figure 7A, please improve the slide resolution of CD163 staining. Whether the radiation/ captopril have the same effect on CD163 expression?
Author Response
Response to the reviewer's comments:
Major concerns:
- Please improve the resolution of all pictures and IHC slides. Most of the figures are difficult to present the differences.
We will improve the method for importing the figures from Corel Draw to the Word file.
- In Figure 1, there was no sham group data which cohort paradelle to compare with radiation treatment.
The Sham group in each panel is indicated by a white circle, at the "O" time point. The sham groups are present for each cell type. The sham blood cell levels are typical for this strain of mice.
- In Figure 2A, was the IHC from radiation + vehicle or radiation + captopril? Please show both IHC staining results in Figure 2A.
The Prussian blue staining for radiation+vehicle was basically the same as for radiation+captopril. The radiation+captopril images are now shown in Supplemental Figure 1 together with radiation+vehicle of the same time point.
- Please label the protein name in the left western blots in figure 4B.
Figure 4B has been improved.
- It is tough to identify the difference between IHC slides in Figure 5. Please improve the resolution of the slides.
A high powered image of Figure 5 is shown in Supplemental Figure 2.
- In Figure 6A, IHC slides are not clear to check the ferritin staining. In Figure 6B, the error bar lines were not equal. Please improve them. Whether the combination of radiation and captopril cause a similar effect on ferritin compared to radiation alone?
Captopril had no effect on Prussian blue staining, and our data indicated that captopril did not affect ferritin gene expression. We did not have sufficient sample to repeat these experiments.
- In Figure 7A, please improve the slide resolution of CD163 staining. Whether the radiation/ captopril have the same effect on CD163 expression?
We did not have sufficient samples to repeat the CD163 expression. Furthermore, since CD163 is regulated post-transcriptionally, we were not able to perform qPCR to obtain this data. We have therefore removed this figure from our manuscript.
Round 2
Reviewer 3 Report
There are no further questions I would like to ask the authors.